

# The prognostic value of tumor length to resectable esophageal squamous cell carcinoma: a retrospective study

Xiangwei Zhang[1], Yang Wang[2], Cheng Li[3], Jing Helmersson[4], Yuanzhu Jiang[1], Guoyuan Ma[1], Guanghui Wang[1], Wei Dong[1], Shaowei Sang[5] and Jiajun Du[1]

[1] Department of Thoracic Surgery, Shandong Provincial Hospital Affiliated to Shandong University, Jinan, China
[2] Department of Medical Imaging, Shandong Provincial Hospital Affiliated to Shandong University, Jinan, China
[3] President's Office, Shandong Cancer Hospital Affiliated to Shandong University, Shandong Academy of Medical Sciences, Jinan, China
[4] Department of Public Health and Clinical Medicine, Epidemiology and Global Health, Umea University, Umea, Sweden
[5] Clinical Epidemiology Unit, Qilu Hospital of Shandong University, Jinan, China

Corresponding authors
Shaowei Sang,
sangshaowei1@163.com
Jiajun Du, dujiajun2011@126.com

## ABSTRACT

**Background**. The current TNM classification system does not consider tumor length for patients with esophageal carcinoma (EC). This study explored the effect of tumor length, in addition to tumor depth and lymph node involvement, on survival in patients with esophageal squamous cell carcinoma (ESCC).

**Methods**. A total of 498 ESCC patients who underwent surgical resection as the primary treatment were selected in the retrospective study. Pathological details were collected, which included tumor type, TNM stage, differentiation. Other collected information were: the types of esophageal resection, ABO blood group, family history and demographic and lifestyle factors. A time-dependent receiver operating characteristic (ROC) curve and a regression tree for survival were used to identify the cut-off point of tumor length, which was 3 cm. Univariate and multivariate Cox proportional hazard regression models were used to identify the prognostic factors to ESCC.

**Results & Discussion**. The 1-, 3-, 5-year overall survival rates were found to be 82.5%, 55.6%, and 35.1%, respectively. Patients who had larger tumor length (>3 cm) had a higher risk for death than the rest patients. From the univariate Cox proportional hazards regression model, the overall survival rate was significantly influenced by the depth of the tumor and lymph node involvement (either as dummy or continuous variables), Sex, and tumor length. Using these four variables in the multivariate Cox proportional hazard regression model, we found that the overall survival was significantly influenced by all variables except Sex. Therefore, in addition to the depth of the tumor and lymph node involvement (as either dummy or continuous variables), the tumor length is also an independent prognostic factor for ESCC. The overall survival rate was higher in a group with smaller tumor length (≤3 cm) than those patients with larger tumor length (>3 cm), no matter what the tumor stage was.

**Conclusion**. The tumor length was found to be an important prognostic factor for ESCC patients without receiving neoadjuvant therapy. The modification of EC staging system may consider tumor length to better predict ESCC survival and identify higher risk patients for postoperative therapy.

## INTRODUCTION

Esophageal cancer is the eighth most common cancer worldwide and the sixth most common cause of death from cancer (*Ferlay et al., 2010*). Esophageal adenocarcinoma (EAC) and squamous cell carcinoma (ESCC) are the predominant histological types. In the western countries, the majority of the esophageal cancers were EAC, while in the eastern countries, ESCC were predominant.

In the current American Joint Committee on Cancer (AJCC) staging system for esophageal tumors, stage depends on the depth of the tumor (T classification), lymph node involvement (N classification), and distant metastasis (M classification), without consideration on tumor length (*Edge & Compton, 2010*). Studies have shown that pathologic tumor length had prognostic value to EAC (*Bolton et al., 2009*; *Gaur et al., 2011*; *Yendamuri et al., 2009*). In China, a few studies were conducted to show the association between the overall survival (OS) rate and the tumor length, among the survival of ESCC patients (*Feng, Huang & Zhao, 2013*; *Ma et al., 2015*; *Wang et al., 2011*; *Wang et al., 2012*). However, the optimal cut-off points for tumor length to predict OS has no consensus. In China, ESCC is the predominant histological subtype and EAC remains rare. Esophageal cancer is the fourth most frequently diagnosed cancer and the fourth leading cause of cancer death in China, with an estimated 259,235 new cases and 211,084 deaths in 2008 (*Lin et al., 2013*). Data from the Chinese National Health and Family Planning Commission databases showed that from 2005 to 2012 the crude incidence and mortality rate for esophageal cancer increased from 19.55 to 38.44/100,000 and 12.62 to 16.77/100,000, respectively (*Tang et al., 2015*). Survival rate from esophageal cancer is low, despite improvements in care. This leads to great disease burden.

More accurate methods of determining prognosis in patients with esophageal cancer are needed. It will benefit patients with specific and appropriate treatment options. The purpose of this study is to explore the prognostic value of pathological tumor length in the context of other prognostic factors and obtain the optimum cut-off point for tumor length.

## MATERIALS & METHODS

### Patients

From August 2007 to July 2014, 552 patients were identified to have undergone surgical resection of esophageal tumor with curative intent at Shandong Provincial Hospital affiliated to Shandong University. The patients who had histology other than squamous cell carcinoma were excluded from this study ($n = 33$). The patients who received neoadjuvant radiotherapy or chemotherapy and noncurative resection (tumor-free margin) were also excluded from this study ($n = 21$). A total of 498 patients with ESCC were included for analyses. The preoperative evaluation was performed based on a complete medical history, physical examination, endoscopic ultrasound and thoracoabdominal computed
tomography scanning. Either the left or the right thoracotomy was included in this study that mainly depended on the tumor location and possible mediastinal lymph nodes involvement. The surgeons taking into account the preoperative evaluation decided the extent of the esophageal resection and lymph nodes dissection during the operation. Then the specimens were sent for pathology examination after preservation in 10% formalin. The tumor length, differentiation, T classification, number of positive and removed lymph nodes were recorded according to the pathology reports. The TNM staging was performed according to the AJCC 7th edition guidelines. Other collected information were: the types of esophageal resection, ABO blood group, family history and demographic and lifestyle factors. After surgery, all the patients were followed up regularly by telephone interviews. Complete follow-up information for all patients until death or July 2014 was available. All subjects gave written informed consent to the study protocol, which was approved by the Ethical Committees of Shandong Provincial Hospital affiliated to Shandong University (No. 2015-055). The study was carried out in accordance with the approved guidelines.

## Statistical analyses

Two methods were conducted to identify the optimum cut-off point for pathologically measured tumor length. First, we used a time-dependent receiver operator characteristic (ROC) curve to evaluate the tumor length in order to predict ESCC survival. Using the ROC curve, we identified the sensitivity and specificity at each tumor length point and the optimal tumor length cut-off point. This was found to be 3.1 cm (Fig. S1A)—based on the highest Youden index (sensitivity + specificity − 1) (*Heagerty, Lumley & Pepe, 2000*). Then, we used a regression tree survival analyses for pathologically measured tumor length (*Gaur et al., 2011*) and found it to be 2.9 cm (Fig. S1B). The mean of these two results - 3.0 cm - was used as the threshold value for optimal cut-off. Based on this value, patients were divided into two groups: one with tumor length ≤3 cm and the other with tumor length >3 cm. Overall survival rate was estimated by the Kaplan–Meier method using the date of esophageal resection as the starting point and the date of death or last follow-up as the endpoint. The Kaplan–Meier survival curves between two tumor length groups were compared using the log-rank test. The association between risk factors and survival was performed using a univariate Cox proportional hazards (Coxph) regression model. Hazard ratios (HRs) with 95% confidence intervals (CIs) were used to quantify the strength of the association. Risk factors with $p$-values ≤0.1 in univariate analyses were entered into a multivariate Coxph regression model. The results of the multivariate Coxph regression model were visualized in graphs. Tests and graphical diagnostics for proportional hazards were based on the scaled Schoenfeld residuals, which calculated tests of the proportional hazards assumption for each covariate by correlating the corresponding set of scaled Schoenfeld residuals with a suitable transformation of time (*Hess, 1995*). All statistical calculations were conducted using R software (version 3.2.3) (*Team, 2014*). Different packages were used for analysis: *survivalROC* package in time-dependent ROC curve, *rpart.plot* and *survival* packages in regression tree for survival, *survival* package in overall survival rate estimated by Kaplan–Meier method and the log-rank test, univariate and

multivariate Coxph regression analyses, tests and graphical diagnostics for proportional hazards.

## RESULTS

The characteristics of the 498 patients are summarized in Table 1. The median follow-up time for all patients was 47.2 months (IQR: 6.5–64.5 months). The median age was 59 years (range 38-81 years); majority of patients were male (78.9%). Over 50% of the patients had over 10 years of smoking or alcohol intake. Over 80% patients' relatives (first-degree) had history of cancer, which includes oral, esophagus, lung, stomach, nasopharynx, liver, pancreatic, rectal, prostate cancer, etc. Majority patients (63%) had tumor with a length larger than 3 cm.

Figure 1 showed the Kaplan–Meier survival curves. The 1-, 3-, 5-year OS rates were 82.5%, 55.6%, and 35.1%, respectively (Fig. 1A). Figure 1B shows a comparison of survival curves between two groups of patients based on their pathologically measured tumor lengths: larger or less and equal to 3 cm. Patients who had larger tumor length showed a lower survival curve than the patients with smaller tumor length. Specifically, the 3-year and 5-year OS were 44.7% and 26.4% for the group with larger tumor length (>3cm); whereas they were 73.1% and 49.3% for the group with smaller tumor length ($\leq$3 cm). This difference is significant ($p < 0.001$)).

Univariate Coxph regression model showed that the following variables had no statistical association with survival: age, ABO blood group, family history of first degree relatives with cancer, types of esophageal resection, histologic grade, length of smoking time and length of alcohol drinking time (either as dummy variable or continuous variable). However, T status and N status, either as dummy variable or continuous variable, and Sex, tumor length influenced significantly the OS (Table 2).

Based on the results from Univariate Coxph regression, a multivariate Coxph regression model was constructed using variables of Sex, T stauts, N status and tumor length. The restults showed that except Sex, the rest three variables have significant association with survival. Therefore, T status and N status (either as dummy variable or continuous variable), and tumor length remained as independent prognostic factors.

Based on the multivariate Coxph regression model, patients with larger tumor length (>3cm) had an increased risk of death compared with patients with smaller tumor length ($\leq$3 cm) after controlling T status and N status. The hazard ratios were 1.52 (95% CI [1.06–2.22]) when treating T and N status as dummy variables and 1.53 (95% CI [1.05–2.21]) when treating T and N status as continuous variables. When T status and N status were treated as continuous variables, the hazard ratios become 1.60 (95% CI [1.21–2.13]) when T status increased by 1 degree after controlling N status and tumor length, and 1.37 (95% CI [1.18–1.60]) when N status increased by 1 degree after controlling T status and tumor length (Table 3).

Figure 2 showed the fitted results of multivariate Coxph regression model. The top four figures showed the variation of T-status (incresing tumor depth) while holding N-status constant at $N_0$; the bottom four figures showed the variation of N-satus (increasing lymph

**Table 1  Characteristics of patients.**

| Characteristic | No. of patients (%) |
| --- | --- |
| **Sex** | |
| Male | 393(78.9) |
| Female | 105(21.1) |
| **Age, y** | |
| Median | 59 |
| Range | 38–81 |
| **ABO blood group** | |
| A | 152(30.5) |
| B | 136(27.3) |
| O | 110(22.1) |
| AB | 64(12.9) |
| Unkown | 36(7.2) |
| **Length of smoking time (year)** | |
| None | 157(31.5) |
| 1–10 | 35(7) |
| 11–20 | 76(15.3) |
| 21–30 | 130(26.1) |
| 30– | 100(20.1) |
| **Length of alcohol drinking time (year)** | |
| None | 211(42.4) |
| 1–10 | 19(3.8) |
| 11–20 | 101(20.3) |
| 21–30 | 100(20.1) |
| 30– | 67(13.5) |
| **Family history in first degree relatives with any cancer** | |
| None | 423(84.9) |
| Yes | 75(15.1) |
| **Types of esophageal resection** | |
| Ivor-Lewis esophagectomy | 143(28.7) |
| Three-field esophagectomy | 47(9.4) |
| Minimally invasive esophagectomy | 17(3.4) |
| Left thoracic | 275(55.2) |
| Unkown | 16(3.2) |
| **Tumor length (cm)** | |
| ≤3 | 185(37.1) |
| >3 | 313(62.9) |
| **Tumor classification** | |
| Tis | 7(1.4) |
| T1 | 47(9.4) |
| T2 | 105(21.1) |
| T3 | 314(63.1) |
| T4 | 21(4.2) |

**Table 1** (*continued*)

| Characteristic | No. of patients (%) |
|---|---|
| Unknown | 4(0.8) |
| **Lymph node classification** | |
| N0 | 294(59) |
| N1 | 102(20.5) |
| N2 | 76(15.3) |
| N3 | 22(4.4) |
| Unknown | 4(0.8) |
| **Grade** | |
| Well differentiated | 16(3.2) |
| Moderately differentiated | 396(79.5) |
| Poorly differentiated | 66(13.3) |
| Undifferentiated | 1(0.2) |
| Unknown | 19(3.8) |

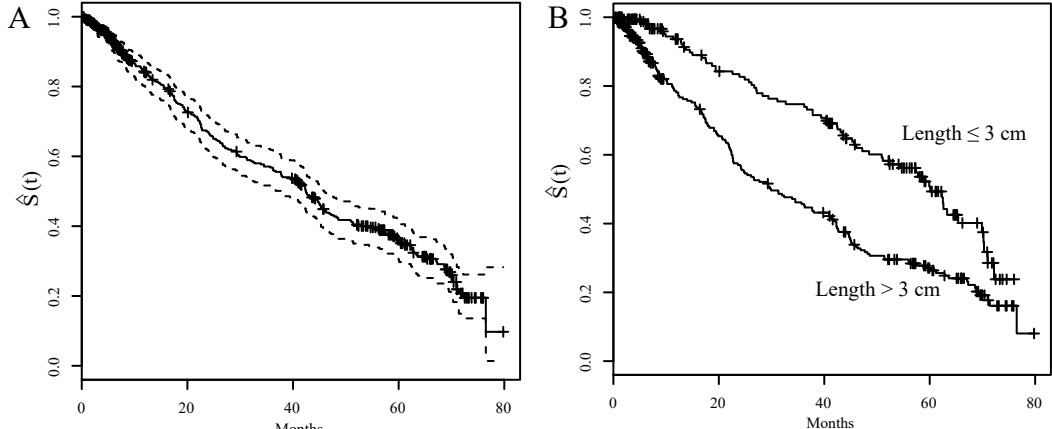

**Figure 1** **Kaplan-Meier survival curves (S(t)).** (A) S(t) vs. time of after surgery with 95% confidence interval for overall ESCC patients. (B) S(t) vs. time of after surgery for ESCC patients stratified by tumor length (3 cm)

node involvement) while keeping T-satus constant at $T_3$. Each figure showed two curves with the top one (blue) corresponding to smaller tumor length and the low curve (green) to larger tumor length. These choices of $N_0$ and $T_3$ are based on the fact that most patients in the study were either in $T_3$ (63.1%) status or in $N_0$ (59%) status (Table 1). We used the multivariable Coxph regression model to fit the survival trend of patients with $T_3N_{0-3}$ status and $T_{1-4}N_0$ status in the two different tumor length groups (tumor length $\leq 3$ cm versus >3 cm). The results showed that the OS rate was higher in the smaller tumor length ($\leq 3$ cm) group independent of the tumor stage. The OS rate declined as either T-status or N-status increases independent of the tumor length.

**Table 2  Univariate Cox proportional regression analyses results.**

| Variable | Univariate analysis | |
|---|---|---|
| | HR (95% CI) | p |
| **Sex** | | |
| Male | 1.00 | |
| Female | 0.64 (0.43–0.94) | 0.023 |
| **Age,y** | 1.00 (0.98–1.02) | 0.836 |
| **Blood type** | | |
| A | 1.00 | |
| B | 1.42 (0.83–2.44) | 0.198 |
| O | 1.41 (0.81–2.45) | 0.225 |
| AB | 1.29 (0.73–2.28) | 0.378 |
| **Length of smoking time (year) (As dummy variable)** | | |
| None | 1.00 | |
| 1–10 | 1.03 (0.57–1.86) | 0.910 |
| 11–20 | 0.95 (0.60–1.50) | 0.818 |
| 21–30 | 0.98 (0.66–1.44) | 0.906 |
| 30– | 0.94 (0.62–1.41) | 0.761 |
| **Length of smoking time (year) (As continuous variable)** | 1.00 (0.99–1.01) | 0.644 |
| **Length of alcohol drinking time (year) (As dummy variable)** | | |
| None | 1.00 | |
| 1–10 | 1.35 (0.65–2.80) | 0.424 |
| 11–20 | 1.11 (0.75–1.64) | 0.593 |
| 21–30 | 1.04 (0.70–1.55) | 0.842 |
| 30– | 1.26 (0.81–1.95) | 0.303 |
| **Length of alcohol drinking time (year) (As continuous variable)** | 1.00 (0.99–1.01) | 0.596 |
| **Family history in first degree relatives with any cancer** | | |
| None | 1.00 | |
| Yes | 0.94 (0.60–1.49) | 0.806 |
| **Types of esophageal resection** | | |
| ILE, MIE, 3-field | 1.00 | |
| Left thoracic | 0.81 (0.60–1.09) | 0.162 |
| **Tumor length (cm)** | | |
| ≤3 | 1.00 | |
| >3 | 2.35 (1.68–3.27) | <0.001 |
| **Tumor classification (As dummy variable)** | 2.14 (1.69–2.75) | |
| $Tis/T_1$ | 1.00 | |
| $T_2$ | 4.94 (1.76–13.82) | <0.001 |
| $T_3$ | 8.45 (3.12–22.87) | <0.001 |
| $T_4$ | 17.93 (5.22–61.54) | <0.001 |
| **Tumor classification (As continuous variable)** | | |

**Table 2** (*continued*)

| Variable | Univariate analysis | |
|---|---|---|
| | HR (95% CI) | *p* |
| **Lymph node classification (As dummy variable)** | | |
| N | 1.00 | |
| $N_1$ | 2.17 (1.51–3.11) | <0.001 |
| $N_2$ | 3.26 (2.23–4.76) | <0.001 |
| $N_3$ | 2.73 (1.45–5.13) | <0.001 |
| **Lymph node classification (As continuous variable)** | | |
| **Grade** | | |
| Well differentiated | 1.00 | |
| Moderately differentiated | 1.07 (0.63–1.82) | 1.823 |
| Poorly differentiated | 0.33 (0.08–1.46) | 1.455 |

**Table 3** **Multivariate Cox proportional regression analyses results.**

| Variable | Dummy variable result | | Continuous variable result | |
|---|---|---|---|---|
| | HR(95% CI) | *p*-value | HR(95% CI) | *p*-value |
| **Sex** | | | | |
| Male | 1.00 | | 1.00 | |
| Female | 0.74 (0.49–1.10) | 0.138 | 0.72 (0.49–1.08) | 0.11 |
| **Tumor length (cm)** | | | | |
| ≤3 | | | | |
| >3 | 1.52 (1.05–2.21) | 0.026 | 1.53 (1.06–2.22) | 0.025 |
| **Tumor classification** | | | 1.60 (1.21–2.13) | <0.001 |
| $Tis/T_1$ | 1.00 | | | |
| $T_2$ | 4.02 (1.43–11.33) | 0.009 | | |
| $T_3$ | 4.82 (1.72–13.50) | 0.003 | | |
| $T_4$ | 7.22 (1.93–27.00) | 0.003 | | |
| **Lymph node classification** | | | 1.37 (1.18–1.60) | <0.001 |
| $N_0$ | 1.00 | | | |
| $N_1$ | 1.67 (1.16–2.42) | 0.006 | | |
| $N_2$ | 2.21 (1.48–3.30) | <0.001 | | |
| $N_3$ | 2.14 (1.13–4.07) | 0.02 | | |

# DISCUSSION

Our study showed that patients with larger tumor length than 3 cm had a higher risk for death than those with smaller tumor length. This indicated that tumor length, in additional to the depth of the tumor and lymph node involvement, is also an independent prognostic factors to ESCC. This finding can be important to the improvement of ESCC prognosis, giving the current poor situation on prognosis of ESCC. In fact, the results in ESCC are inconsistent so far. The worldwide studies have been trying to identify more accurate methods to determine prognosis in patients. This study provides the needed evidence for the prognostic factors to improve the ESCC.
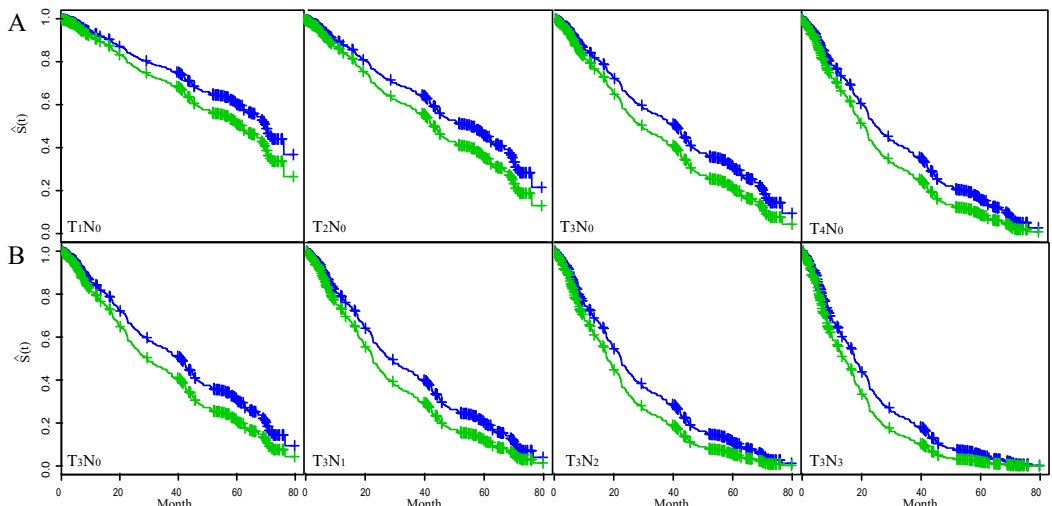

**Figure 2** **The fitted result of ESCC patients based on the multivariate Cox proportional hazard regression model.** The fitted survival curve (S(t)) for ESCC patients with $T_{1-4}N_0$ status (A) and $T_3N_{0-3}$ status (B). Each figure showed S(t) for two different tumor length groups. Blue line: tumor length ≤3 cm; Green line: tumor length >3 cm.

The association between tumor length and OS rate has been controversial. Two studies conducted in Japan and one study in Germany showed that tumor length was associated with survival rate of ESCC patients in univariate analyses, but not in the multivariate analyses (*Bollschweiler et al., 2006*; *Tachibana et al., 2002*; *Tachibana et al., 1999*). In these studies, some patients received neoadjuvant therapy, which may have biased the measurement of tumor length because of tumor regression. Another study conducted in Japan also showed that tumor length was not an independent prognostic factor to OS of ESCC patients (*Igaki et al., 2001*). However, the study sample was small—only 96 ESCC cases. In addition, the tumor length was measured with X-ray examination which is known to be a rough measuring method. Further more, the study did not indicate how to figure out the cut-off point. A study conducted in America showed that tumor length had no association with survival of ESCC patients (*Yendamuri et al., 2009*). Here again, the statistical power was small because of small sample size—only 30 cases.

Some studies conducted in China showed that tumor length could be as a prognostic factor in ESCC (*Feng, Huang & Zhao, 2013*; *Ma et al., 2015*; *Wang et al., 2011*). However, the optimum cut-off point of the tumor length was different. One study in Taibei indicated that 3 cm was the cut-off point for prediction (*Wang et al., 2011*); while the other study in Tianjin and the third one in Zhejiang chosed 4 cm as the cut-off point (*Feng, Huang & Zhao, 2013*; *Ma et al., 2015*). A large study conducted in the US showed that tumor length was an independent predictor of mortality and a 3-cm tumor length cut-off was proposed. This agrees with our result using regression tree for survival and time-dependent ROC curve methods, and multivariate Coxph regression.

Our study indicated that T classification and N classification both were independent prognostic factors in EC. This is consistent with the known results that are acknowledged

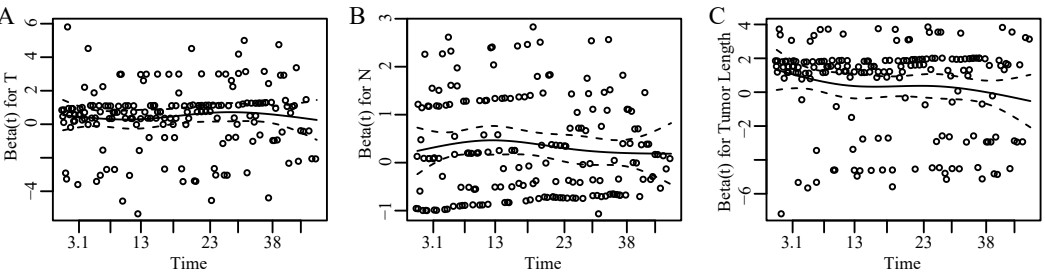

**Figure 3** **Scaled Schoenfeld residuals for each covariate.** The residual was estimated as the time-dependent coefficient beta(t) vs. transformed time. The solid line is a smoothing spline fit to the plot and the broken lines represent a ±2-standard-error band around the fit. The assumption of proportional hazards was supported for each of the covariates with no trend (horizontal line centering around 0) and $p > 0.05$ (Table S1). (A–C) represent results of scaled Schoenfeld residuals against transformed time for T, N and tumor length, respectively.

in the current TNM classification (*Peyre et al., 2008*; *Wijnhoven et al., 2007*). Fig. 3 showed the dependence of Beta for T, N, and tumor length on time. Beta was the scaled Schoenfeld residuals from tests and graphical diagnostics for proportional hazards. If the proportional hazards assumption is true, beta(t) will be a horizontal line centering around 0. Since Beta for each chosen variable was independent of time and basically zero, we concluded that each prognostic factor and the whole model did not violate the proportional hazards assumption (Table S1). Our results showed that the survival rate decreased as either tumor depth or lymph node involvement increased after controlling the other variables. The multivariable Coxph regression model indicated that tumor length was positively associated with survival after adjusting the T and N status. Therefore, the optimal cut-off point of tumor length applied to different T or N status.

Studies have proven that smoking and alcohol intake increased the risk of ESCC (*Chung et al., 2010*; *Islami et al., 2011*), and positive synergistic effect of alcohol and tobacco use for ESCC existed (*Prabhu, Obi & Rubenstein, 2014*). Our study explored whether smoking and alcohol intake before diagnosis had influence on ESCC survival. However, the study showed that alcohol and tobacco intake had no association with survival. The data on smoking and alcohol were consumption duration not the intensity of tobacco and the amount of alcohol usage, which was an important limitation. One study showed that heavy smoking is strongly associated with poor prognosis (*Shitara et al., 2010*).

In our study, we did not find the association between ESCC and either family history of any cancer in first-degree relatives or family history of EC in first-degree relatives. Very few studies explore the prognostic value of family history of cancer to ESCC. Other studies have shown that family history of any cancer in first-degree relatives was one of the risk factor for ESCC (*Bhat et al., 2015*; *Gao et al., 2009*), and *Chen et al. (2015)* proved that family history of EC was the risk factor of ESCC. In addition, Jiang et al. indicated that family history of EC was one of the independent prognostic factors for survival of ESCC (*Yuequan, Shifeng & Bing, 2010*). While there was no obvious evidence to confirm ESCC with positive family history was a hereditary predisposition (*Yuequan, Shifeng & Bing, 2010*). More studies are needed to confirm this association.

In the study, we did not find the association between ABO blood group and survival. This is a controversial area. Several studies explored the ability of ABO blood type to predict prognosis of ESCC, but the results were not consistent. *Wang et al. (2015)* showed that ABO blood group had no association with EC survival. *Qin et al. (2015)* indicated that blood type was not an independent prognostic factor for all ESCC patients, but was an independent prognostic factor for ESCC patients with negative lymph nodes. *Yang, Huang & Feng (2014)* found that ABO blood group was an independent prognostic factors and blood group O had a worse OS than non-O blood groups. The conflicting results could be due to the heterogeneity of study populations or limited study sizes. Although there are several hypotheses which may explain the relationships observed between ABO blood group and cancer, the direct biologic mechanisms underlying the association are inconclusive. Therefore, whether ABO blood group can be used to predict ESCC survival or not still needs to be confirmed.

Our study indicated that tumor length was a prognostic factor for ESCC patients. However, limitations still existed in our study. The study design was a retrospective observational study. As known, generally the evidence level of retrospective study is relatively low compared to that of prospective study. In addition, the sample size of the study was relatively small, which introduced the relatively larger 95% CI of HR (Table 3). In the future, we plan to conduct a multi-center prospective study to verify and quantity the results more precisely.

## CONCLUSIONS

In this retrospective study of 498 patients, we have found that tumor length was an important prognostic factor for ESCC patients without receiving neoadjuvant therapy, in addition to the tumor depth and lymph node involvement. As with lung cancer and breast cancer, tumor size may be considered in the modification of EC staging system to better predict ESCC survival and identify high risk patients for postoperative therapy. Further prospective study with larger sample may be needed to confirm our results.

### Funding
This work was supported by the Medical and Health Technology Development Program of Shandong Province (2016WS0432). The funders had no role in study design, data collection and analysis, decision to publish, or preparation of the manuscript.

### Grant Disclosures
The following grant information was disclosed by the authors:
Medical and Health Technology Development Program of Shandong Province: 2016WS0432.

### Competing Interests
The authors declare there are no competing interests.

## Author Contributions

- Xiangwei Zhang conceived and designed the experiments, analyzed the data, wrote the paper, prepared figures and/or tables.
- Yang Wang conceived and designed the experiments, analyzed the data, prepared figures and/or tables.
- Cheng Li, Yuanzhu Jiang, Guoyuan Ma, Guanghui Wang and Wei Dong contributed reagents/materials/analysis tools.
- Jing Helmersson contributed reagents/materials/analysis tools, reviewed drafts of the paper.
- Shaowei Sang and Jiajun Du conceived and designed the experiments, reviewed drafts of the paper.

## Human Ethics

The following information was supplied relating to ethical approvals (i.e., approving body and any reference numbers):

Ethical Committees of Shandong Provincial Hospital affiliated to Shandong University (No. 2015-055).

## Data Availability

The raw data has been supplied as a Supplementary File.

## Supplemental Information

Supplemental information for this article can be found online at http://dx.doi.org/10.7717/peerj.2943#supplemental-information.

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
