# Peer review of "The prognostic value of tumor length to resectable esophageal squamous cell carcinoma: a retrospective study"

_PeerJ, doi:10.7717/peerj.2943_

## Round 0.1 · original submission · Major Revisions

Your manuscript has been reviewed and found to be of potential acceptance by some reviewers. However, the manuscript is not acceptable in its present form for publication in PeerJ. The reviewers hae delineated several deficiencies and recommend major revisions which may render th work suitable for publication.

Please review the peer review comments carefully and edit the manuscript to respond to them. Please attach to your revised manuscript a point-by-point response to the reviewer's comments along with an explanation of any request of the reviewers that you do NOT address in your revised manuscript.

Sincerely,

Prof. Cheorl-Ho Kim, Ph.D
Editor
The Peer Journal

Reviewer 1 ·

Basic reporting

In the current study, 498 ESCC patients who underwent surgical resection as the primary treatment were selected to evaulate the value the prognostic value of tumor length using univariate and multivariate Cox proportional hazard regression models. The result is that the patients who had larger tumor length (>3 cm) had a higher risk fo death than the rest patients. The authors conclude that the tumor length is an important prognostic factor for ESCC patients without receiving neoadjuvant therapy.

Experimental design

Retrospective study; all patients were not treated by neoadjuvant therapy; univariate and multivariate Cox proportional hazard regression models

Validity of the findings

Although the tumor length appears to be predictive for survival in this study, there is no independent validation cohort to confirm the findings. Moreover, most patients with ESCC receive neoadjuvant treatment in these days and the effect of the neoadjuvant therapy had not been investigated in this study. Nevertheless, this study contributes to the field of ESCC and all findings are appropriatly stated. This study should be intresting for the readers.

Additional comments

In the current reprospective study the authors state that the tumor length is predictive for survival in patients with ESCC. However, there is no independent validation cohort to confirm the findings. Moreover, most patients with ESCC receive neoadjuvant treatment in these days and the effect of the neoadjuvant therapy had not been investigated in this study. Nevertheless, this study contributes to the field of ESCC and all findings are appropriatly stated.

Reviewer 2 ·

Basic reporting

The authors should explain the detail for each figure.

Experimental design

The authors should explain the detail for methods.

Validity of the findings

No comments.

Additional comments

In the current study, the authors conclude that the tumor length is an important prognostic factor for patients with ESCC. The study was interesting, however, the authors have not add something new to the literature. There are many studies regarding tumor length for patients with ESCC.
1. In the current, the authors used two methods to identify the optimal cutoff value for tumor length (ROC curve and regression tree). The authors should explain the detail how to determinate the cutoff point by these two methods. The authors should add the figures for these two methods, and indicate the sensitivity and specificity for these two methods.
2. The authors confirmed that 3cm as the optimal value. The authors should add TNM stage in the current study. Can this value apply to the different TNM stage.
3. The authors should add the survival and p-value for figure 2.
4. The authors should explain the detail for figure 3.

---

## Round 0.2 · accepted · Accept

Thank you for your careful revision with incorporation of the criticisms.
Although your study is limited in the sample size, the trial study on the prognostic factors associated with survival is interesting in further generalization as a retrospective study. Therefore, the present Editor responsible for your manuscript would like to accept to publish in the PeerJ.

We will also thank you for your prospective study triggered from the present work.

Cheorl-Ho Kim
Editor